# Experimental and Theoretical Study on Carbonization Coefficient Model of NS/SAP Concrete

**Shiquan Li** [1,2] , **Yu Chen** [2,*] , **Can Tang** [2] , **Jinyuan Wang** [2] , **Ronggui Liu** [2,3] **and Haojie Wang** [4]

1. Taizhou Institute of Science and Technology, Nanjing University of Science and Technology, Taizhou 225300, China
2. Faculty of Civil Engineering and Mechanics, Jiangsu University, Zhenjiang 212013, China
3. School of Civil Engineering and Architecture, Nantong Institute of Technology, Nantong 226000, China
4. Tianping College of Suzhou University of Science and Technology, Suzhou 215009, China
* Correspondence: chenyu_ujs@163.com

**Abstract:** Carbonization coefficient research has great significance in concrete carbonization evaluation. Nano-silica (NS) can reduce the content of $Ca(OH)_2$, which is generated during the hydration of concrete, resulting in improved carbonization resistance and compressive strength of concrete. This paper investigates the carbonization effects of concrete with internal curing, such as Super Absorbent Polymer (SAP). The research shows that SAP can promote hydration of the internal concrete but form tiny pores after releasing the water completely, which may cause a reduction of carbonation resistance of concrete. The concrete was modified by adding SAP, ranging from 0 to 0.24%, to ascertain the optimal content of SAP. The addition of NS changed the concrete from 0 to 1.5% to confirm the optimal range of NS. To establish a reasonable suitable theoretical model of NS/SAP concrete, the influence factors of the carbonization coefficient of concrete were analyzed first. Later, the accelerated carbonization test was carried out on 100 mm × 100 mm × 100 mm cube specimens with different carbonization time to obtain the compressive strength and carbonization depth to establish the carbonization coefficient model of NS/SAP concrete. Before analyzing experimental data, the specimens were randomly divided into fitting and validation groups. Based on the regression analysis of the fitting group, the carbonization coefficient model was established, which embodied the influence of various parameters on concrete carbonization, including SAP content, NS content, water–cement ratio, $CO_2$ concentration, temperature, relative humidity, and compressive strength. According to the validation analysis of the verification group, the mean relative error of the model is 5.04%, and the residual mean square error is 0.1751. Compared with the literature models, this study's carbonization model can accurately predict the carbonization depth of NS/SAP concrete.

**Keywords:** concrete carbonization; accelerated carbonization; carbonization model; internal curing; super absorbent polymer (SAP); nano-silica (NS)

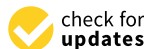



## 1. Introduction

The carbonization of concrete will reduce the alkalinity of concrete gradually and lead to concrete deterioration and corrosion of steel bar, which is one of the fundamental indicators for the durability evaluation of reinforced concrete structures [1–3]. Relevant scholars have established concrete carbonization prediction models based on different factors, which are suitable for different kinds of concrete [4]. The carbonization models are mainly of the following types: (1) diffusion theory model of $CO_2$, including the Alekseev model [5] and the Papadakis model [6]; (2) empirical model of water–cement ratio, including the Koichi Kishitani model [7], Akihiko Ida model [8], Zhu An-min model [9], and Uomoto Kenichi model [10]; (3) multi-factor empirical model, including Richardson model [10], Gong Luoshu model [11], Huang Shi-yuan model [12], and Zhang Hai-yan model [13]; (4) empirical model based on compressive strength, including Smolczyk model [8], Di Xiao-tan model [11], Soviet model [14], and Niu Ditao model [15]; (5) semi-theoretical and

semi-empirical model, including Zhang Yu model [16], Liu Yaqin model [17], and "1 + 2" carbonization model [18]. The above models show that the effects of concrete curing are crucial to the carbonization resistance and compressive strength of concrete.

Curing measures of concrete are divided into external and internal, while the common external curing measures including watering, spraying curing agent, and surface covering, which usually lead to insufficient hydration inside the concrete. Considering the curing of the internal concrete, water-absorbent materials such as Super Absorbent Polymer (SAP) can be incorporated into the concrete. During the curing period, the SAP inside the concrete can release water gradually, which will enhance the overall hydration degree and increase the matrix compactness, thus resisting the transmission of $CO_2$.

However, some research [19,20] has shown that using internal curing material may weaken the carbonization resistance of concrete. The internal curing effect of SAP will produce more $Ca(OH)_2$, which can reduce the carbonization resistance activity of concrete. Therefore, specific measures need to be adopted to prevent the degradation of carbonation resistance of the internal curing concrete.

Nano-silica (NS) has extremely high volcanic ash activity [21] and can be used in conjunction with SAP in concrete. The volcanic ash effect of NS will consume the $Ca(OH)_2$ generated by the secondary hydration of SAP, which can reduce the negative effect of SAP on the carbonization and improve the carbonization resistance of concrete ultimately. However, the theoretical analysis of NS modification on the carbonization resistance of SAP concrete needs further study. Therefore, it is necessary to analyze the carbonization resistance of NS/SAP concrete and explore its carbonization model.

In this paper, the influencing factors of carbonization of NS/SAP concrete were analyzed. After the theoretical analysis, the carbonization coefficient theoretical model of NS/SAP concrete was established preliminarily. According to the accelerated carbonization test data of NS/SAP concrete, the model parameters were clarified and verified. Compared with the literature, the carbonization coefficient theoretical model of NS/SAP concrete has better applicability.

## 2. Experimental Framework

### 2.1. Materials and Apparatus

P.O. 42.5 Portland cement (Chinese standard GB 175-2007/XG3-2018 [22]) is the main cement-based material. Parameters of SAP(class II, China building materials industry standard, JC/T 2551-2019 [23]) and hydrophilic NS(type A, Chinese standard GB/T 20020-2013 [24]) are shown in Tables 1 and 2, respectively. Crushed stone with particle size range of 5 mm to 20 mm (Chinese standard GB/T 14685-2022 [25]) was used as the coarse aggregate. River sand with fineness modulus of 2.7 (Chinese standard GB/T 14684-2022 [26]) was used as the fine aggregate. Polycarboxylate water reducer (CQJ-JSS type, Chinese standard GB 50119-2013 [27]) was used to improve the workability of concrete. Water for cement hydration (Chinese industry standard JGJ 63-2006 [28]) was potable water. Phenolphthalein indicator solution (10 g/L, Chinese standard GB/T 603-2002 [29]) and paraffin wax (Chinese standard GB 1886.26-2016 [30]) were prepared for concrete carbonization test.

**Table 1.** Material parameters of SAP.

| Type | Particle Diameter (mm) | Density (g/mL) | Water Absorption (g/g) | PH Value |
|------|------------------------|----------------|------------------------|----------|
| II | 0.2 | 0.65 to 0.85 | 400 to 500 | 5.5 to 6.5 |

**Table 2.** Material parameters of NS.

| Particle Diameter (nm) | $SiO_2$ Content (%) | Specific Surface Area ($m^2$/g) | PH Value |
|------------------------|---------------------|----------------------------------|----------|
| 20 | 99.5 | 200 | 4.0 to 7.0 |

The main apparatus involved: (1) HJW-200 Single-horizontal-shaft forced type concrete mixer; (2) HZJ-A concrete vibrating stand; (3) CCB-70 carbonization test chamber; (4) SYE-

2000 compression testing machine; (5) KX-1613T ultrasonic vibration meter; (6) SHBY-90B concrete curing box; (7) QZ-9425AE industry drying oven; and (8) HT-1 carbonation depth measuring instrument, etc.

### 2.2. Specimen Fabrication and Testing

### 2.2.1. Mix Proportions

In the research by Hu [31], when the content of SAP is less than 0.5%, the self-shrinkage of concrete can be significantly reduced, and the loss of compressive strength is not significant. In this research, the SAP content included 0, 0.08%, 0.16%, and 0.24%, respectively. The NS content included 0, 0.5%, 1.0%, and 1.5%, respectively. The mixture proportions of concretes list in Table 3. In Table 3, "S16N5" represents the content of SAP and NS are 0.16% and 0.5%, respectively. The content of SAP and NS is calculated according to the total mass of the cementitious material, which includes cement, SAP, and NS. The content of polycarboxylate water reducer was 0.20% of the total mass of the cementitious material. The additional water consumption was determined according to the Powers model [32], as shown in Equation (1).

$$\frac{W_e}{C} = \begin{cases} 0.18\frac{W}{C}, & \text{for } \frac{W}{C} \leq 0.36 \\ 0.42 - \frac{W}{C}, & \text{for } 0.36 < \frac{W}{C} \leq 0.42 \end{cases} \tag{1}$$

where $W$ is the mass of water for cement hydration, $C$ the mass of cement, $W_e$ the mass of extra water for internal curing, $W/C$ the mass ratio of water to cement, and $W_e/C$ the mass ratio of extra water to cement. In this research, $W/C = 0.35$. Thus, $W_e/C = 0.063$ according to Equation (1).

**Table 3.** Mix proportions of concrete specimens (kg/m$^3$).

| Mix Ratio Number | Cement | River Sand | Crushed Stone | Total Water | Water for Hydration | Extra Water | SAP | NS | Water Reducer |
|---|---|---|---|---|---|---|---|---|---|
| S0N0 | 520.00 | 680 | 1020 | 182.00 | 182.00 | 0 | 0 | 0 | 1.040 |
| S8N0 | 519.58 | 680 | 1020 | 214.59 | 181.85 | 32.73 | 0.42 | 0 | 1.039 |
| S16N0 | 519.17 | 680 | 1020 | 214.42 | 181.71 | 32.71 | 0.83 | 0 | 1.038 |
| S24N0 | 518.75 | 680 | 1020 | 214.24 | 181.56 | 32.68 | 1.25 | 0 | 1.038 |
| S0N5 | 517.40 | 680 | 1020 | 181.09 | 181.09 | 0 | 0 | 2.6 | 1.035 |
| S0N10 | 514.80 | 680 | 1020 | 180.18 | 180.18 | 0 | 0 | 5.2 | 1.030 |
| S0N15 | 512.20 | 680 | 1020 | 179.27 | 179.27 | 0 | 0 | 7.8 | 1.024 |
| S8N5 | 516.98 | 680 | 1020 | 213.51 | 180.94 | 32.57 | 0.42 | 2.6 | 1.034 |
| S8N10 | 514.38 | 680 | 1020 | 212.44 | 180.03 | 32.41 | 0.42 | 5.2 | 1.029 |
| S8N15 | 511.78 | 680 | 1020 | 211.37 | 179.12 | 32.24 | 0.42 | 7.8 | 1.024 |
| S16N5 | 516.57 | 680 | 1020 | 213.34 | 180.80 | 32.54 | 0.83 | 2.6 | 1.033 |
| S16N10 | 513.97 | 680 | 1020 | 212.27 | 179.89 | 32.38 | 0.83 | 5.2 | 1.028 |
| S16N15 | 511.37 | 680 | 1020 | 213.20 | 178.98 | 32.22 | 0.83 | 7.8 | 1.023 |
| S24N5 | 516.15 | 680 | 1020 | 213.17 | 180.65 | 32.52 | 1.25 | 2.6 | 1.032 |
| S24N10 | 513.55 | 680 | 1020 | 212.10 | 179.74 | 32.35 | 1.25 | 5.2 | 1.027 |

Note: "S": SAP; "N": NS; $W/C = 0.35$; $M_{\text{cementitious material}} : M_{\text{stone}} : M_{\text{sand}} = 26:51:34$.

### 2.2.2. Preparation of Concrete Specimens

All the specimens were prepared according to Chinese Standard GB/T 50082-2019 [33] (Standard for test methods of concrete physical and mechanical properties) and GB/T 50081-2009 [34] (Standard for Test Methods of Long-term Performance and Durability of Ordinary Concrete).

NS/SAP concrete specimens with a size of 100 mm × 100 mm × 100 mm for each mix proportion were casted in the laboratory. The step-by-step process of concrete specimen preparation is depicted in Figure 1. All specimens were cured with mold at room temperature for 24 h, and then stripped and cured at $T = 20 \pm 2$ °C and $RH \geq 95\%$ in the curing box until 28 days [33].

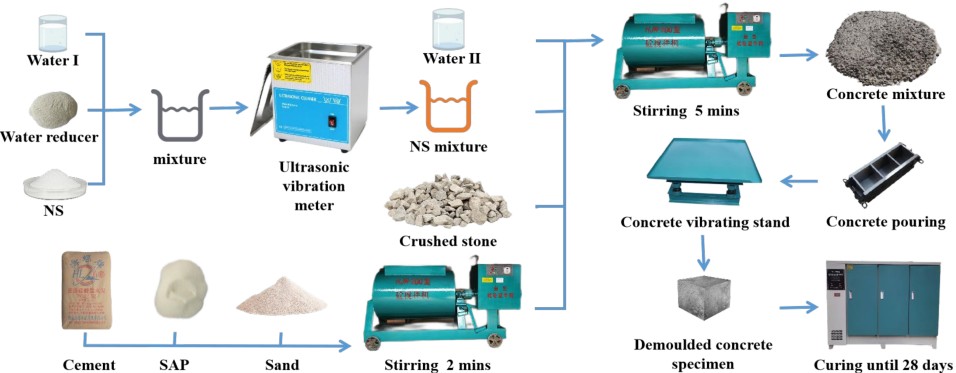

**Figure 1.** Preparation procedure and process of concrete specimens.

### 2.2.3. Accelerated Carbonization Tests

All specimens placed in the oven dry at 60 °C for 48 h after curing for 28 days in Concrete Curing Box. Then, specimens were coated with paraffin wax on four successive sides and placed in a carbonization chamber to accelerate carbonization at $20 \pm 5$ °C, while relative humidity was $70\% \pm 5\%$ and $CO_2$ concentration was $20\% \pm 3\%$. When the predetermined carbonization time (3 d, 7 d, 14 d, and 28 d) was reached, the compressive strength and carbonization depth of specimens were tested [34]. The steps of the carbonization depth test include: (1) split the middle of the carbonized surface of the specimen, (2) spray the phenolphthalein alcoholic indicator on the cross sections, (3) measure the carbonization depth at intervals of 10 mm along the edge of the cross section; giving the result accurate to 0.5 mm. The process of compressive and concrete carbonization test is shown in Figure 2.

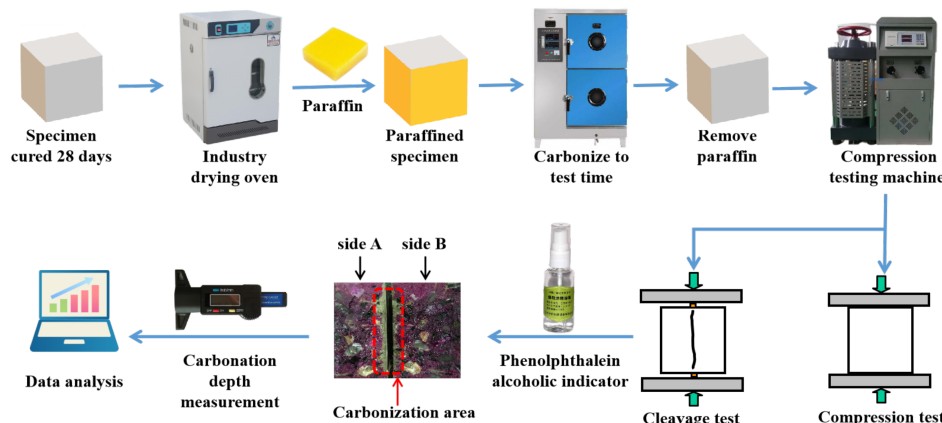

**Figure 2.** Procedure for carbonization area test of concrete specimens.

### 3. Theoretical Model Analysis

Steps for establishing a multivariate model [35] include: (1) determine the prediction object and target; (2) select the appropriate regression method; (3) establish a sample set from the experimental data; (4) determine parameters and establish a model; (5) analyze prediction of the model; and (6) correct the model.

The corresponding steps in this study include: (1) analyze the relationship between variables and carbonization coefficient, (2) clarify the mathematical relationship of variables and carbonization coefficient, (3) superimpose the mathematical relationship between variables and carbonization and establish the overall model, (4) carry out the coefficient fitting, and (5) verify and analyze the model.

The expression of concrete carbonization depth as given by Equation (2):

$$X = K \cdot \sqrt{\frac{t}{365}} \tag{2}$$

where $X$ is carbonization depth, mm; $K$ carbonization coefficient; $t$ carbonization time, day.

According to Equation (2), it is obvious that the carbonation depth of concrete relates to the carbonation coefficient. This study will be devoted to the theoretical analysis of the concrete carbonization coefficient.

### 3.1. Analysis of External Influencing Factors

#### 3.1.1. $CO_2$ Concentration

The higher the external $CO_2$ concentration $k_c$, the greater the gradient difference between the internal and external of concrete, which leads to an increase in carbonization coefficient $K$. According to the Alekseev model [5], Zhang Yu model [16], and Liu Ya-qin model [17], $K$ can be taken as proportional function of $k_c^{0.5}$.

#### 3.1.2. Relative Humidity

The prerequisite for concrete carbonization is that $H_2O$ forms carbonic acid in combination of $CO_2$. The relative humidity ($RH$) of curing environment will affect the saturation of pore water in concrete. With decrease of $RH$, the inside of the concrete will be much drier. In this condition, although the diffusion rate of $CO_2$ increases, the carbonation reaction will slow down due to the lack of water. With the increase of $RH$, the pore water saturation of concrete will increase. The water generated by the carbonation reaction is hard to release, which weakens the diffusion of $CO_2$ and eventually leads to inhibit the carbonation reaction.

Ji [36] found that the carbonization coefficient conforms to the parabolic relationship with $RH$. The carbonation rate of concrete will improve, when $RH$ is 40–70%. Du [37] proposed that the carbonization rate was the fastest when $RH$ was 50–60%. According to GB/T 50082 [33], the $RH$ of concrete accelerated carbonization test in this study is 70%.

Jiang [38] statistically analyzed literature data and presented the influence of $RH$ on concrete carbonation conformed to Equation (3).

$$\frac{k_{RH_1}}{k_{RH_2}} = \frac{(1 - RH_1)^{1.1}}{(1 - RH_2)^{1.1}} \tag{3}$$

where $RH_i$ is the relative humidity of $i$-th environment; $k_{RHi}$ the carbonization rate under the relative humidity $RH_i$.

According to the carbonization model proposed by Niu [15], Liu [17], and Jiang [38], $K$ is taken to be the quadratic of $RH$.

#### 3.1.3. Ambient Temperature

The ambient temperature $T$ has a great influence on the concrete carbonization. The increase of $T$ promotes the diffusion of $CO_2$ and enhances the carbonization rate. However, the increase of $T$ also decreases the solubility of $CO_2$, which weakens the carbonization rate [39]. Loo [40] concluded that the effect of $T$ is not significant from 20 °C to 40 °C. However, it has also been suggested that the carbonization reaction intensifies during the increase of $T$ from 20 °C to 40 °C when $RH$ is constant. According to the carbonization database [38], the effect of $T$ on the carbonization rate is shown in Equation (4).

$$\frac{k_{T_1}}{k_{T_2}} = \left(\frac{T_1}{T_2}\right)^{0.25} \tag{4}$$

where $T_j$ is the $j$-th ambient temperature, °C; $k_{Ti}$ the carbonization rate when the temperature is $T_i$.

As is exhibited in Equation (5), the effect of temperature on the carbonization rate is obtained by regression analysis of Uomoto K. [11].

$$k_T = e^{(8.748 - \frac{256.3}{T+273.15})}$$ (5)

According to Uomoto Kenichi model [11] and Niu Di-tao model [15], $K$ is taken to be a power of $T$.

### 3.2. Analysis of Internal Influencing Factors

#### 3.2.1. Water–Cement Ratio

With the increase of water–cement ratio $W/C$, the internal porosity of concrete increases. The pores may be the diffusion channel of $CO_2$, leading to the increase of carbonation rate. With the decrease of water–cement ratio $W/C$, concrete is denser, and the carbonation rate will reduce.

According to Koichi Kishitani model [7] and Uomoto Kenichi model [11], carbonization depth increases with $W/C$. In this paper, the carbonation coefficient $K$ is considered to proportional to $W/C$.

#### 3.2.2. Compressive Strength of Concrete

As is known to all, compressive strength of concrete $f_{cu,k}$ is the comprehensive result of the mix ratio, materials, curing conditions and other factors. However, in the Smolczyk model [8] and Niu Di-tao model [15], $f_{cu,k}$ was proposed as a parameter of carbonization coefficient $K$, which was a power of $f_{cu,k}$.

#### 3.2.3. The Content of SAP and NS

In the carbonization test of this section, 35 groups of specimens were tested, each group with 3 samples. According to the carbonization test of specimens with SAP or NS, Figures 3 and 4 were obtained. It can be observed from Figure 3 that $X$ showed an increasing trend with the increase of $R_{SAP}$, which is consistent with the reference [19] and $X$ showed a gradual increase with increasing age of carbonization. During the water release process of SAP, discontinuous micro-pores form inside the concrete, which provides additional possibilities for the diffusion and infiltration of $CO_2$ in the later stage. Therefore, the depth of concrete carbonization showed an increasing trend with the increase of SAP content $R_{SAP}$. As shown in Figure 4, with the increase of $R_{NS}$, $X$ decreases first and then increases, and the optimal value of $R_{NS}$ is 1.0%. Appropriate addition of NS can make the micro-structure of concrete more compacted, improving the carbonization resistance. According to the above analysis, the carbonization coefficient $K$ is taken as proportional to the quadratic of $R_{NS}$ and proportional to $R_{SAP}$.

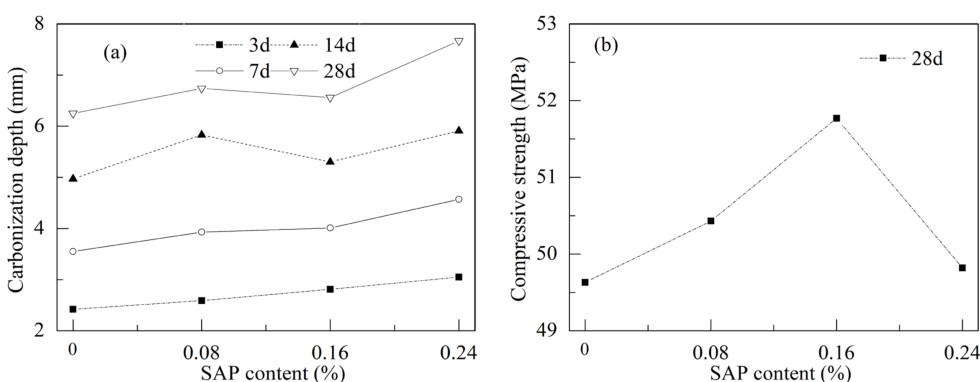

**Figure 3.** SAP concrete with different carbonization time: (**a**) Carbonization depth vs. SAP content; (**b**) compressive strength vs. SAP content.

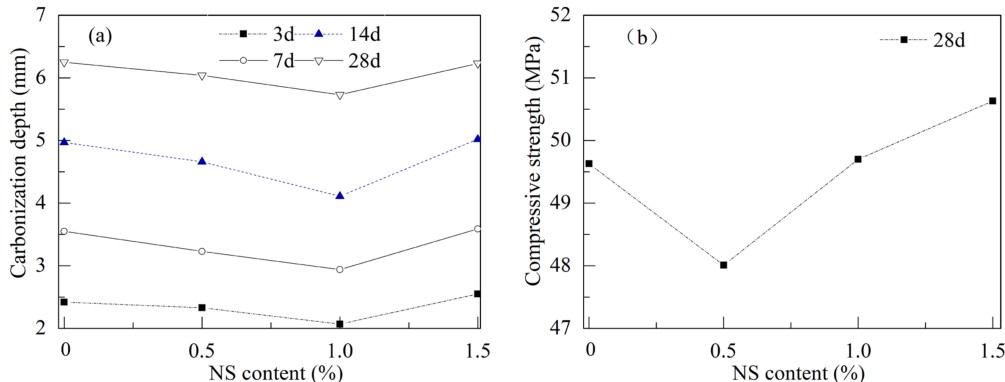

**Figure 4.** NS concrete with different carbonization time: (**a**) Carbonization depth vs. NS content; (**b**) compressive strength vs. NS content.

### *3.3. Theoretical Model*

The concrete carbonization coefficient $K$ can be taken as the dependent variable. $K$ can be taken as power-squared with $f_{cu,k}$, $k_c$, and $T$, respectively, with a linear relationship with $W/C$ and $R_{SAP}$, proportional to the quadratic of $RH$ and $R_{NS}$. The multivariate model of $K$ established as Equation (6).

$$K = \alpha_1 f_{cu,k}^{\beta_1} + \alpha_2 T^{\beta_2} + \alpha_3(RH^2 + \beta_3 RH) + \alpha_4(R_{NS}^2 + \beta_4 R_{NS}) + \alpha_5 \frac{W}{C} + \alpha_6 k_c^{0.5} + \alpha_7 R_{SAP} + \alpha_8 \qquad (6)$$

where $\alpha_i$ is the regression coefficient, $\beta_j$ the power coefficient.

## 4. Theory Model Building and Validation

In the carbonization test of this section, 36 other groups of specimens were tested, each group with 3 samples. All the carbonization depth tests and compressive strength tests were carried out on the specimens after accelerated carbonization for 28 days. The results of specimens of 25 groups as shown in Table 4. Randomly, take 30% of 25 groups as the verification group (recorded as No.1 to No.8), and the rest of the groups as the fitting group.

The results of the fitting group were used for coefficient fitting in the theoretical model as in Equation (6). The carbonization depth of the verification group calculated by Equation (6) will compare with the test results of the verification group.

### *4.1. Specimen Analysis*

As shown in Figures 5 and 6, the two-factor coordinate diagram was established according to Table 4, which shows the relationship between the carbonization depth of the specimens and the content of NS or SAP. The carbonization depth of the unadulterated concrete (as the reference specimen) was 6.25 mm as shown in Figure 5. The carbonization depth of the single SAP specimens is higher, while the highest carbonization depth is 7.67 mm when the content of SAP is 0.24%. For different contents of SAP, the addition of NS will reduce the carbonization depth of the specimens. The carbonization depth decreases first and then increases with the increase of NS content, and the effect is best when the NS content is 1.0%. Overall, when the content of NS and SAP is 1.0% and 0.16%, respectively, the carbonization depth of the re-doped specimen is 5.30 mm, which is the smallest. The carbonization depth reduced by 15.2% compared with the reference specimen. The volcanic ash effect of NS is obvious, which can improve the carbonization resistance of SAP concrete.

**Table 4.** Specimen parameters and carbonization depth test value.

| Group No. | Mix Ratio Number | SAP Content (%) | NS Content (%) | Carbonization Depth (mm) | | Compressive Strength (MPa) | |
|---|---|---|---|---|---|---|---|
| | | | | Mean | Standard Deviation | Mean | Standard Deviation |
| 1 | S0N0 | 0 | 0 | 6.25 | 0.18 | 49.63 | 1.39 |
| 10 | S0N5 | 0 | 0.5 | 6.04 | 0.17 | 48.01 | 1.02 |
| 11 | S0N10 | 0 | 1.0 | 5.73 | 0.24 | 49.70 | 1.75 |
| 7 | S0N15 | 0 | 1.5 | 6.23 | 0.34 | 50.63 | 1.48 |
| 8 | S8N0 | 0.08 | 0 | 6.74 | 0.57 | 50.43 | 1.50 |
| 12 | S8N5 | 0.08 | 0.5 | 6.49 | 0.32 | 48.49 | 2.46 |
| 13 | S8N10 | 0.08 | 1.0 | 6.12 | 0.34 | 46.91 | 0.66 |
| 14 | S8N15 | 0.08 | 1.5 | 6.69 | 0.31 | 49.17 | 1.60 |
| 21 | S8N5 | 0.08 | 0.5 | 6.46 | 0.28 | 48.39 | 0.96 |
| 22 | S8N10 | 0.08 | 1.0 | 6.11 | 0.15 | 49.67 | 1.67 |
| 23 | S8N15 | 0.08 | 1.5 | 6.60 | 0.36 | 47.72 | 2.06 |
| 9 | S16N0 | 0.16 | 0 | 6.56 | 0.35 | 51.77 | 1.54 |
| 15 | S16N5 | 0.16 | 0.5 | 6.81 | 0.38 | 52.08 | 1.86 |
| 16 | S16N10 | 0.16 | 1.0 | 5.28 | 0.38 | 47.28 | 0.81 |
| 17 | S16N15 | 0.16 | 1.5 | 6.34 | 0.43 | 45.85 | 1.65 |
| 24 | S16N5 | 0.16 | 0.5 | 6.78 | 0.41 | 49.43 | 1.53 |
| 25 | S16N10 | 0.16 | 1.0 | 5.31 | 0.29 | 47.28 | 1.38 |
| 3 | S16N15 | 0.16 | 1.5 | 6.66 | 0.31 | 48.26 | 1.80 |
| 2 | S24N0 | 0.24 | 0 | 7.67 | 0.21 | 49.82 | 1.18 |
| 4 | S24N5 | 0.24 | 0.5 | 7.53 | 0.35 | 46.90 | 0.97 |
| 5 | S24N10 | 0.24 | 1.0 | 6.13 | 0.20 | 48.63 | 1.45 |
| 6 | S24N15 | 0.24 | 1.5 | 6.95 | 0.42 | 47.11 | 1.66 |
| 18 | S24N5 | 0.24 | 0.5 | 7.44 | 0.45 | 47.29 | 1.66 |
| 19 | S24N10 | 0.24 | 1.0 | 6.25 | 0.26 | 48.75 | 2.12 |
| 20 | S24N15 | 0.24 | 1.5 | 6.83 | 0.11 | 46.47 | 0.94 |

Note: Group No.1 to No.8 as the verification group, No.9 to No.25 as the fitting group.

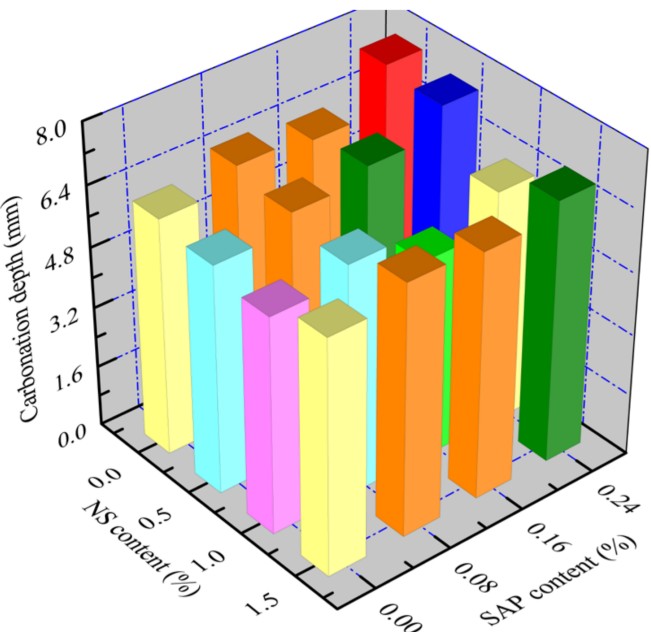

**Figure 5.** Carbonization depth of NS/SAP concrete (carbonization for 28 days).

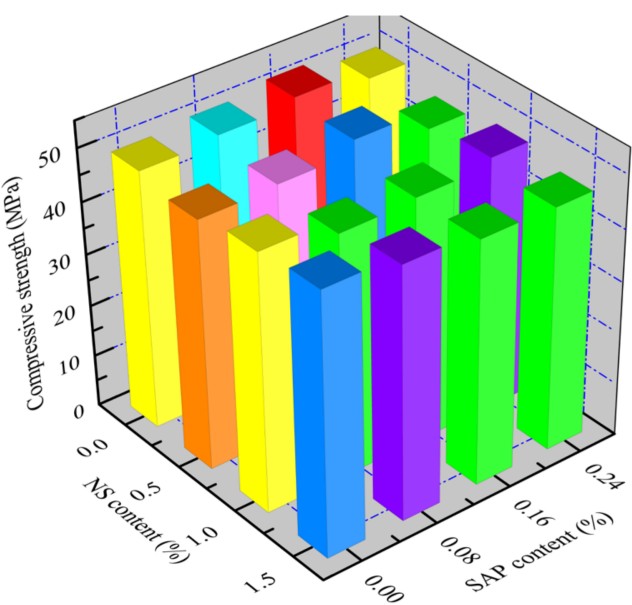

**Figure 6.** Compressive strength of NS/SAP concrete (carbonization for 28 days).

As shown in Figure 6, the compressive strength of benchmark specimen is 49.63 MPa. The strength of the single-doped SAP specimens is higher than that of the benchmark specimen. The highest compressive strength of concrete is 51.77 MPa when the SAP dose is 0.16%. The strength of the single-doped NS specimens improved with the increase of the NS content. The highest compressive strength of specimens is 50.63 MPa when NS content is 1.5%. This is because NS is a nanoscale material and exerts a micro-aggregate effect in concrete. The average strength of all composite specimens is 47.96 MPa, which is 3.4% less than the reference specimens. This indicates that SAP slightly reduces the strength of the composite specimens after 28 days. However, the internal curing effect of SAP is mainly in the continuous improvement of the concrete strength in the later period.

### 4.2. Carbonization Model of Compounded Specimens

According to the test data of the fitted groups (No.9 to No.25) in Table 4, Equation (2) and the parameters of the test conditions (water–cement ratio is 0.35, $T$ is 20 $°C$, $RH$ is 70%, and $CO_2$ concentration is 20%), the coefficients $\alpha_i$ and $\beta_i$ are determined by fitting a multivariate nonlinear regression through the nonlinear fitting function in *Matlab*. Substituting $\alpha_i$ and $\beta_i$ into Equation (6), the multivariate nonlinear model of the carbonization coefficient $K$ of the NS/SAP concrete is obtained, as shown in Equation (7).

$$
\begin{aligned}
K = &-0.1492 f_{\text{cu,k}}^{0.1832} + 0.9666 T^{0.4888} - 2.1226(RH^2 + 1.1042RH) \\
&+0.0053(R_{\text{NS}}^2 - 10.094 R_{\text{NS}}) + 1.5771\tfrac{W}{C} - 2.331 k_{\text{c}}^{0.5} + 0.0907 R_{\text{SAP}} + 0.552
\end{aligned}
\tag{7}
$$

### 4.3. Model Validation and Analysis

The compressive strength, SAP content, and NS content of the verification group in Table 4 substitute into Equations (7) and (2) to obtain the predicted carbonization depth of the verification specimens, as shown in Table 5 and Figure 7. According to Table 5, (1) the overall mean value of the relative error is 5.04%; (2) the residual mean square deviation of this model is 0.1751.

**Table 5.** Prediction and analysis of carbonization depth of specimens.

| Group No. | Carbonization Depth (mm) | | | Relative Error (%) |
|---|---|---|---|---|
| | Test Value | Predicted Value | Error | |
| 1 | 6.25 | 6.65 | 0.40 | 6.40 |
| 2 | 7.67 | 7.24 | −0.43 | −5.61 |
| 3 | 6.66 | 6.55 | −0.11 | −1.65 |
| 4 | 7.53 | 6.72 | −0.81 | −10.76 |
| 5 | 6.13 | 6.54 | 0.41 | 6.69 |
| 6 | 6.95 | 6.76 | −0.19 | −2.73 |
| 7 | 6.23 | 6.14 | −0.09 | −1.44 |
| 8 | 6.74 | 6.65 | −0.09 | −1.34 |

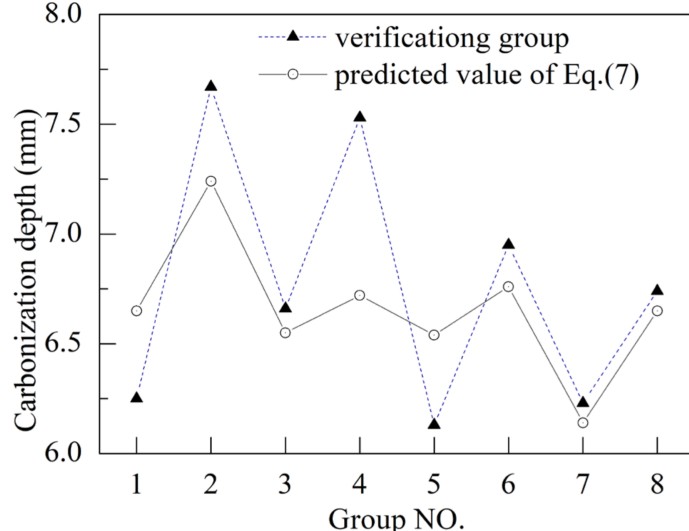

**Figure 7.** Comparison and prediction of carbonization depth.

Table 6 shows the prediction error analysis of carbonization depth of concrete with different admixture according to the models in the existing research [41–43] and this paper. Compared with reference [41–43], the number of groups in this paper is suitable. Table 6 exhibits that the relative error between the predicted value and the measured value of the model in this paper is small, and the accuracy of the model is better than that of the related studies.

**Table 6.** Carbonization model analysis of different concrete.

| Literature | Concrete Admixture | Number of Groups | Relative Error (%) |
|---|---|---|---|
| Reference [41] | None | 16 | 6 |
| | FA | 49 | 12 |
| | SG | 13 | 13 |
| Reference [42] | FA/SG/SF | 27 | 5.47 |
| Reference [43] | RA | 12 | 18 |
| | RA/BF | 12 | 0 |
| | BF | 12 | 19 |
| this research | NA/SAP | 25 | 5.04 |

Note: FA: fly ash; SF: silica fume; SG: slag; BF: basalt fiber; RA: recycled aggregate.

In general, the prediction model with a smaller residual error mean square deviation is more accurate. Some studies analyzed the relative error and residual mean square

error at the same time according to the predicted value and the measured value from their respective studies. The relative error is 16% and the residual mean square error is 0.34 in Reference [44]; the relative error is 23% and the residual mean square error is 0.43 in Reference [45]. The residual mean square error of this model is 0.1751, which has better accuracy.

## 5. Conclusions

In this paper, based on the theoretical models, the influence factors of concrete carbonization were analyzed. According to the experimental data, the NS/SAP concrete carbonization model is established and analyzed. The main conclusions are summarized as follows:

The multivariate nonlinear model can predict the carbonization depth of NS/SAP concrete. The relationship between various factors and the carbonization coefficient were analyzed to establish the corresponding mathematical relationship and the carbonization coefficient model. Combined with the existing theoretical models, the relationship between the carbonization coefficient of NS/SAP concrete can be clarified as: (1) proportional to the power of $CO_2$ concentration, (2) proportional to the quadratic of relative humidity, (3) proportional to the power of ambient temperature, (4) proportional to the water–cement ratio, (5) proportional to the power of concrete compressive strength, (6) proportional to the SAP content, and (7) proportional to the quadratic of NS content.

As internal curing material, the addition of SAP can improve the carbonization resistance of concrete, but slightly weakens its strength in the short term. The micro-aggregate effect and volcanic ash effect of NS are obvious, which can improve the strength of SAP concrete. The results of the orthogonal experiment show that, when the content of NS and SAP are 1.0% and 0.16%, respectively, the carbonization resistance of the compound specimens will enhance in evidence, while the compressive strength of NS/SAP concrete is 47.28 MPa.

The coefficient of the theoretical model was clarified by nonlinear fitting based on the actual measured values of carbonization depth of the specimens. The mean relative error between the predicted values and the experimental values is 5.04%, and the residual mean square error is 0.1751. Compared with similar studies, the resulting carbonization coefficient model of NS/SAP involved more influencing factors and has good applicability and accuracy.

**Author Contributions:** Conceptualization, S.L. and J.W.; data curation, S.L., J.W. and R.L.; formal analysis, S.L. and Y.C.; funding acquisition, S.L., C.T., Y.C. and R.L.; investigation, S.L., J.W. and H.W.; project administration, R.L.; resources, R.L. and Y.C.; supervision, R.L. and Y.C., validation, S.L. and J.W.; writing—original draft, S.L. and H.W.; writing—review and editing, S.L., C.T. and Y.C. All authors have read and agreed to the published version of the manuscript.

**Funding:** This research was funded by the National Natural Science Foundation of China (No.51508234, No.51778272), Natural Science Foundation of Science and Technology Department of Jiangsu Province (No.BK20180878), and Jiangsu Provincial Department of Education ("Qinglan Project" of 2022).

**Institutional Review Board Statement:** Not applicable.

**Informed Consent Statement:** Not applicable.

**Data Availability Statement:** Data are contained within the article.

**Acknowledgments:** The authors gratefully acknowledge the financial support from the National Natural Science Foundation of China (No.51508234, No.51778272), Natural Science Foundation of Science and Technology Department of Jiangsu Province (No.BK20180878), and Jiangsu Provincial Department of Education ("Qinglan Project" of 2022).

**Conflicts of Interest:** The authors declare that they have no known competing financial interest or personal relationships that could have appeared to influence the work reported in this paper.

**Abbreviations**

| Symbol | Interpretation |
|---|---|
| $W$ | mass of water for cement hydration |
| $W_e$ | mass of extra water for internal curing |
| $C$ | mass of cement |
| $W/C$ | mass ratio of water to cement |
| $W_e/C$ | mass ratio of extra water to cement |
| $M$ | mass of material |
| $X$ | carbonization depth |
| $K$ | carbonization coefficient |
| $t$ | carbonization time |
| $k_c$ | external $CO_2$ concentration |
| $k_{RH}$ | carbonization rate under $RH$ |
| $T$ | ambient temperature |
| $k_T$ | carbonization rate of $T$ |
| $f_{cu,k}$ | compressive strength of concrete |
| $R_{NS}$ | the content of NS |
| $R_{SAP}$ | the content of SAP |
| $\alpha$ | the regression coefficient |
| $\beta$ | the power coefficient |

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
