# Peer review of "Experimental and Theoretical Study on Carbonization Coefficient Model of NS/SAP Concrete"

_buildings, doi:10.3390/buildings12122227_

Round 1
Reviewer 1 Report
The authors must carefully check all the text and improve English grammar.
There is a roughly edited abstract.
The carbonization coefficient research has great significance in concrete carbonization evaluation. Nano-silica (NS) can reduce the content of Ca(OH)2, which generate during the hydration of concrete, resulting improve carbonization resistance and compressive strength of concrete. This paper investigates the carbonization effects of concrete with internal curings, such as Super Absorbent Polymer (SAP). The research shows that SAP can promote hydration of the interior concrete but form tiny pores after releasing the water completely, which may cause a reduction of carbonation resistance of concrete. The concrete was modified by adding SAP, ranging from 0 to 0.24%, to ascertain the optimal content of SAP. The addition of NS changed the concrete from 0 to 1.5% to confirm the optimal range of NS. To establish a reasonable suiTable theoretical model of NS/SAP concrete, the influence factors of the carbonization coefficient of concrete are analyzed first. Later, the accelerated carbonization test was carried out on 100 mm × 100 mm × 100 mm cube specimens with different carbonization times to obtain the compressive strength and carbonization depth to establish the carbonization coefficient model of NS/SAP concrete. Before analyzing experimental data, the specimens were randomly divided into fitting and validation groups. Based on the regression analysis of the fitting group, the carbonization coefficient model was established, which embodied the influence of various parameters on concrete carbonization, including SAP content, NS content, water-cement ratio, CO2 concentration, temperature, relative humidity and compressive strength. According to the validation analysis of the verification group, the mean relative error of the model is 5.04%, and the residual mean square error is 0.1751. Compared with the literature models, this study's carbonization model can accurately predict the carbonization depth of NS/SAP concrete.
The authors must modify the keywords.
- However, some research[19,20] have shown that the use of internal curing material 61 may weaken the carbonization resistance of concrete.
- However, some research[19,20] has shown that using internal curing material 61 may weaken the carbonization resistance of concrete.
- The authors do not cite the Buildings journal.
Author Response
Dear reviewers,
Thank you for the time and effort that they have put into reviewing the previous version of the manuscript. We were pleased to know that our work was rated as potentially acceptable for publication in Journal, subject to adequate revision. The suggestions have enabled us to improve our work.
Based on the instructions provided in your Review Report, we uploaded the file of the revised manuscript. Accordingly, we have uploaded a copy of the original manuscript with all the changes highlighted by using the track changes mode in MS Word.
Appended to this letter is our point-by-point response to the comments raised by the reviewers.
We would like also to thank you for allowing us to resubmit a revised copy of the manuscript.We hope that the revised manuscript is accepted for publication in the Journal of Mountain Science.
Kind regards,
Yu Chen

Reviewer 2 Report
The manuscript entitled “Experimental and Theoretical Study on Carbonization Coefficient Model of NS/SAP Concrete” presents an interesting topic for academics and engineers. However, the manuscript is not suitable for publication. Some need to be revised.
1 - The manuscript has several typing errors;
2 - In figures 3b and 4b, on the Y axis, what is Compressive Strength/mm? That's right?
3 - In table 4 and throughout the text, the units must be placed between parentheses or square brackets, according to the journal model. The "/" symbol represents the division operation!
SAP content/%
NS content/%
Carbonization depth/mm
Compressive strength/MPa
The representations above are not usual!
4 - Insert the number of specimens for each test performed. Insert standard deviation or the coefficient of variation in the presentation of results.
5 - Despite figures 5 and 6 apparently being very didactic, the results of the Z axis are difficult to identify. I suggest inserting a more intuitive color scale or presenting the results in another clearer way.
6 - The most important point that deserves more attention is the analysis of results. The results need to be scrutinized and not just presented. The discussion of the results needs to be more in-depth. The use of the literature is necessary in the analysis of the results.
Author Response

(The authors gave the same response as above.)

Round 2
Reviewer 2 Report
The authors followed the reviewer's suggestions.